# Roles of β-Cell Hypoxia in the Progression of Type 2 Diabetes

**DOI:** 10.3390/ijms25084186

**Published:** 2024-04-10

**Authors:** Kazuya Yamagata, Tomonori Tsuyama, Yoshifumi Sato

**Affiliations:** 1Department of Medical Biochemistry, Faculty of Life Sciences, Kumamoto University, Kumamoto 860-8556, Japan; ysato413@kumamoto-u.ac.jp; 2Center for Metabolic Regulation of Healthy Aging (CMHA), Faculty of Life Sciences, Kumamoto University, Kumamoto 860-8556, Japan; t-tsuyama@kumamoto-u.ac.jp

**Keywords:** hypoxia, pancreatic β-cells, glucotoxicity, type 2 diabetes, hypoxia-inducible factor, transcriptional repressor

## Abstract

Type 2 diabetes is a chronic disease marked by hyperglycemia; impaired insulin secretion by pancreatic β-cells is a hallmark of this disease. Recent studies have shown that hypoxia occurs in the β-cells of patients with type 2 diabetes and hypoxia, in turn, contributes to the insulin secretion defect and β-cell loss through various mechanisms, including the activation of hypoxia-inducible factors, induction of transcriptional repressors, and activation of AMP-activated protein kinase. This review focuses on advances in our understanding of the contribution of β-cell hypoxia to the development of β-cell dysfunction in type 2 diabetes. A better understanding of β-cell hypoxia might be useful in the development of new strategies for treating type 2 diabetes.

## 1. Introduction

Diabetes mellitus, a chronic condition marked by hyperglycemia, is one of the leading causes of death and disability worldwide. It is estimated that 529 million people were living with diabetes globally in 2021, with type 2 diabetes accounting for 96.0% of all cases, and the number of people with diabetes is projected to more than double to 1.3 billion people worldwide by 2050 [1]. Type 2 diabetes results from the complex interplay of multiple genetic and environmental factors. The genetic background causes insulin resistance and β-cell dysfunction, while being overweight and physical inactivity exacerbates these metabolic abnormalities [2,3,4]. Impaired insulin secretion and insulin resistance are characteristic features of type 2 diabetes [2]. In insulin resistance, pancreatic β-cells increase insulin secretion to maintain normal glucose tolerance; however, when β-cells are incapable of increasing insulin secretion, the plasma concentration of glucose increases. Prolonged exposure to hyperglycemia has deleterious effects on β-cell number and function, a concept known as glucotoxicity, and this leads to the development and progression of type 2 diabetes [5,6,7]. Hyperglycemia exerts its toxic effects through various mechanisms, including oxidative stress, endoplasmic reticulum (ER) stress, and inflammation [8,9,10]. However, recent studies have indicated that hyperglycemia also induces hypoxia in β-cells [11,12,13]. Hypoxia, in turn, contributes to β-cell dysfunction via several mechanisms, including the activation of the hypoxia-inducible factor (HIF) pathway [14,15]. Here, we review the current knowledge on β-cell hypoxia, focusing on impaired insulin secretion in type 2 diabetes. A better understanding of β-cell hypoxia might be useful in the development of new strategies for treating type 2 diabetes.

## 2. Induction of Hypoxia in Pancreatic β-Cells by Hyperglycemia

In normoxic pancreatic β-cells, glucose is metabolized into pyruvate via glycolysis and is further oxidized in the mitochondria to produce adenosine triphosphate (ATP) via oxidative phosphorylation. An increase in ATP closes ATP-sensitive potassium channels, leading to membrane depolarization, calcium influx, and the exocytosis of insulin [16]. Cellular oxygen levels are regulated by the balance between the supply and demand of oxygen, and hypoxia occurs when oxygen consumption exceeds the oxygen supply. Given the high demand of mitochondrial oxidative phosphorylation during insulin secretion, β-cells consume large amounts of oxygen. Indeed, we, and others, have demonstrated that pancreatic islets and β-cell lines readily become hypoxic under high-glucose conditions [11,12,13,17]. These studies have also shown that the islets in animal models of type 2 diabetes are hypoxic [11,12,13]. Thus, β-cell hypoxia occurs in vivo. In addition, there is a decrease in blood flow in the islets of animal models of type 2 diabetes [18]. Therefore, insufficient oxygen supply might also be involved in β-cell hypoxia in vivo.

The oxygen tension of most mammalian cells is in the range of 20–65 mmHg (equivalent to 3–9% O_2_) [19] and the mean tissue oxygen tension at the surface of normal mouse islets is 44.7–45.7 mmHg (6.3–6.4% O_2_) [20]. Hypoxic responses are reported to occur at 0.5–5% oxygen tension in culture conditions in vitro [21]. Consistently, exposure of MIN6 β-cells to 5% oxygen tension causes cellular hypoxia with impaired insulin secretion and inhibits β-cell growth; 3% oxygen tension readily induces apoptosis, indicating that moderate hypoxia is a stress factor for β-cells and reduces β-cell number and function [22,23]. Therefore, hypoxic stress is a likely mechanism that underlies β-cell failure in type 2 diabetes [22,24,25] (Figure 1).

## 3. Roles of HIFs in Pancreatic β-Cells

The maintenance of oxygen homeostasis is important for ATP production and energy availability in cells. Therefore, all mammals have processes to sense, respond to, and correct hypoxia. HIFs are the key regulators of oxygen homeostasis in the cellular response to hypoxia. HIFs are members of the basic helix-loop-helix Per-Arnt-Sim transcription factor family and consist of an oxygen-sensitive HIF-α subunit and a constitutively expressed HIF-1β/aryl hydrocarbon receptor nuclear translocator (ARNT) subunit [21,26,27]. There are three forms of HIF-α (HIF-1α, HIF-2α, and HIF-3α), but the majority of HIF transcriptional responses seem to be attributed to HIF-1α and HIF-2α [28]. During normoxic conditions, HIF-α is hydroxylated at the two proline residues within the oxygen-dependent degradation domain by prolyl hydroxylase domain (PHD) proteins in the presence of oxygen, 2-oxoglutarate, and iron. Hydroxylated HIF-α subunits are polyubiquitylated by von Hippel–Lindau protein and targeted for proteasomal degradation. Under hypoxic conditions, HIF-α subunits are prevented from hydroxylation by PHD proteins and subsequent degradation. As a result, stabilized HIF-α dimerizes with HIF-1β and activates a large number of target genes, including those involved in glycolysis, erythropoiesis, and angiogenesis by binding to the hypoxia response elements in their promoter regions.

Three forms of PHD proteins (PHD1, PHD2, and PHD3) are expressed in β-cells [29] and HIF-1α is degraded rapidly under normal oxygen conditions. However, HIF-1α is present in normoxic β-cells [30]. Glucose transporter type 2 (GLUT2) is a low-affinity glucose transporter that is required for the maintenance of normal glucose-stimulated insulin secretion in β-cells [31]. Glucokinase (GCK), a rate limiting enzyme of glycolysis, acts as a glucose sensor for physiological insulin secretion in β-cells [32]. Interestingly, deletion of the *Hif1a* gene in β-cells causes impaired insulin secretion and glucose intolerance in mice with a decreased expression of *Slc2a2* (encoding GLUT2) and *Gck* (encoding glucokinase) [30]. Consistently, HIF-1α knockdown decreases *Slc2a2* and *Gck* expression levels and markedly suppresses insulin secretion in MIN6 β-cells under normoxic conditions [30]. Thus, HIF-1α expression at basal levels is essential for insulin secretion, although the underlying mechanisms for the decreased expression of *Slc2a2* and *Gck* by HIF-1α deficiency are unclear (Figure 2A,B). Furthermore, HIF-1α protects against β-cell destruction in type 1 diabetes, the autoimmune type of diabetes [33,34]. In addition, HIF-1β/ARNT deficiency also impairs insulin secretion by β-cells [35,36]. Intriguingly, reduced HIF-1α and HIF-1β/ARNT expression has been observed in the islets of type 2 diabetic patients [30,35]. Furthermore, HIF-1 signaling is complexly repressed by hyperglycemia through PHD protein-dependent mechanisms [15,37]. These observations strongly suggest that HIF-1 proteins play an important role in maintaining β-cell function, and the impairment of HIF-1 signaling is involved in β-cell dysfunction in type 2 diabetes.

In contrast, it has also been reported that HIF-1 expression is increased in the β-cells of various diabetic animals, including ob/ob mice, mice fed a high-fat diet, and db/db mice [12,38]. Sustained HIF-1 overexpression in β-cells by the deletion of the *Vhl* gene (encoding the von Hippel–Lindau protein) causes impaired insulin secretion and glucose intolerance in mice [39,40,41], indicating that the upregulation of HIF-1α is deleterious to β-cell function and contributes to diabetes. HIF-1 activates the transcription of genes encoding glucose transporter type 1 (GLUT1), glycolytic enzymes (e.g., glucose-6-phosphate isomerase and phosphoglycerate mutase 1), pyruvate dehydrogenase kinase 1 (PDK1), lactate dehydrogenase A (LDHA), and monocarboxylate transporter 4 (MCT4) [42]. PDK1 inactivates pyruvate dehydrogenase, the enzyme that converts pyruvate into acetyl-CoA for the mitochondrial tricarboxylic acid cycle. LDHA stops the entry of pyruvate into the tricarboxylic acid cycle by converting it into lactate, and MCT4 promotes the extrusion of lactate from cells. Consequently, the main impact of HIF-1 in glucose metabolism is a shift in energy metabolism from mitochondrial respiration to glycolysis. However, mitochondrial oxidative metabolism plays a critical role in the control of insulin secretion [43]. The deleterious effects of HIF-1α on insulin secretion might be explained by the attenuation of mitochondrial activity (Figure 2C). Intriguingly, the treatment of diabetic mice with the HIF-1α inhibitor PX-478 improves insulin secretion and glucose tolerance [38], suggesting that the inhibition of HIF-1α might be a potential treatment for type 2 diabetes (Figure 2D). Taken together, these results indicate that a balanced and adequate amount of HIF-1 activity is necessary for normal insulin secretion by pancreatic β-cells.

HIF-2α, a paralog of HIF-1α, also dimerizes with HIF-1β to activate target genes in response to hypoxia. However, HIF-1α and HIF-2α play distinct roles in β-cells. As described above, β-cell-specific *Hif1a* knockout mice exhibit impaired insulin secretion and glucose intolerance [30]. In contrast, HIF-2α deficiency in β-cells does not impair insulin secretion or glucose tolerance in mice on a normal chow diet [44]. A chronic increase in mitochondrial metabolism enhances electron flux along the mitochondrial transport chain, resulting in an increased production of reactive oxygen species (ROS) [44,45,46]. HIF-2α plays important roles in the regulation of cellular redox state by activating anti-oxidant gene expression, including *Sod2* (encoding superoxide dismutase 2) and *Cat* (encoding catalase), and protecting against mitochondrial damage by ROS [47]. Consistently, the expression of anti-oxidant genes is decreased in the islets of β-cell-specific *Hif2a* knockout mice and these mice develop impaired insulin secretion and glucose intolerance when fed a high-fat diet [44]. These results indicate that HIF-2α preserves β-cell function under metabolic overload conditions by stimulating anti-oxidant gene expression.

## 4. Roles of Transcriptional Repressors in Hypoxic β-Cells

HIFs function mainly as transcriptional activators; however, transcriptional repression also occurs to inhibit energy-demanding processes under hypoxic conditions [48]. Indeed, approximately 5% of genes, including some involved in insulin secretion, are downregulated in hypoxic islets and MIN6 β-cells [22,23,49], indicating that gene repression is another important adaptive response to hypoxia in β-cells. Global gene expression analysis revealed that basic helix-loop-helix family member E40 (BHLHE40) and activating transcription factor 3 (ATF3) are hypoxia-induced transcriptional repressors in pancreatic β-cells (Figure 3) [23].

BHLHE40 (also referred to as DEC1/SHARP2/STRA13) is a member of the basic helix-loop-helix family and functions as a transcriptional repressor by binding to DNA at class B E-box motifs [50,51]. The transcription factor musculoaponeurotic fibrosarcoma oncogene family A (MAFA) plays a critical role in glucose-stimulated insulin secretion by regulating the expression of genes involved in insulin exocytosis, including *Stxbp1* (encoding MUNC18-1) and *Stx1a* (encoding syntaxin 1A) [52,53]. Peroxisome proliferator-activated receptor-γ coactivator 1α (PGC-1α), which is encoded by *Ppargc1a*, regulates mitochondrial biogenesis and ATP production [54]. Expression of the transcriptional repressor BHLHE40 is highly induced in β-cells by hypoxia and suppresses insulin secretion by repressing the expression of *Mafa* and *Ppargc1a*. Consistently, β-cell-specific *Bhlhe40* deficiency improves insulin secretion and glucose intolerance in ob/ob mice.

ATF3 also suppresses the expression of the genes involved in glucose metabolism, including *Ins1* (encoding insulin-1), *Ins2* (encoding insulin-2), and *Irs2* (encoding insulin receptor substrate 2) [23,55,56]. Furthermore, the hypoxia-induced upregulation of the pro-inflammatory *Il1b* and pro-apoptotic *Noxa* genes, as well as the activation of caspase-3, are suppressed by *Atf3* deficiency in MIN6 β-cells [23,56,57]. These findings also indicate that the transcriptional repressor ATF3 is involved in hypoxia-induced β-cell dysfunction and loss.

## 5. Hypoxia Regulates Several Stress Pathways in β-Cells

5′-Adenosine monophosphate (AMP)-activated protein kinase (AMPK) is an evolutionarily conserved serine/threonine kinase. AMPK is activated in response to energy stresses, such as hypoxia, by sensing the increase in AMP and/or the adenosine diphosphate/ATP ratio in cells and restores the energy balance by inhibiting anabolic processes that consume ATP, while promoting catabolic processes that generate ATP [58,59]. Hepatocyte nuclear factor 4α (HNF4α), a transcription factor belonging to the nuclear receptor superfamily, plays a critical role in insulin secretion [60,61]. We found that the hypoxia-induced activation of AMPK reduces insulin secretion by reducing the stability of HNF4α [62]. Therefore, downregulation of HNF4α by AMPK activation might be involved in impaired insulin secretion under hypoxic conditions.

Impaired protein homeostasis (termed proteostasis) in the ER leads to the accumulation of unfolded and abnormally folded proteins, called ER stress, which activates the ER unfolded protein response (UPR^ER^) to mitigate proteotoxic stress [63,64]. Hypoxia increases β-cell death by inhibiting the expression of adaptive UPR^ER^ genes, including *Hspa5* (encoding heat shock protein family A member 5), *Hsp90b1* (encoding heat shock protein 90 beta family member 1), *Fkbp11* (encoding FKBP prolyl isomerase 11), and spliced *Xbp1* (encoding X-box binding protein 1). These inhibitory effects of hypoxia are mediated by the activation of c-Jun N-terminal kinase and DNA damage-inducible transcript 3, but are independent of HIF-1α [65]. Inactivation of UPR^ER^ could be a cellular mechanism for increased cell death by hypoxic stress.

Oxidative stress is provoked in various tissues under high-glucose conditions. Of note, β-cells are particularly vulnerable to ROS due to their low expression levels of antioxidant enzymes, including catalase, glutathione peroxidase, and mitochondrial manganese superoxide dismutase, and ROS produced in β-cells reduce insulin gene expression by decreasing the expression and/or DNA binding activity of the pancreatic and duodenal homeobox 1 (PDX1) transcription factor [66,67]. Intriguingly, hypoxia also increases ROS generation at the mitochondrial electron transport chain [68,69]. These results strongly suggest that hypoxia-induced ROS production is also involved in β-cell dysfunction.

From the above, hypoxia affects multiple steps during the process of glucose-stimulated insulin secretion. Specifically, hypoxia attenuates insulin secretion by shifting glucose metabolism from mitochondrial respiration to glycolysis through the activation of HIF-1. Hypoxia also inhibits insulin secretion by suppressing the expression of MAFA (exocytosis) and PGC-1α (ATP production) through the activation of transcriptional repressor BHLHE40. In addition, the hypoxia-induced activation of AMPK downregulates the expression of HNF4α, leading to defective insulin secretion. Furthermore, hypoxia-induced ROS production might inhibit insulin gene expression through the dysregulation of PDX1 (Figure 4).

## 6. Conclusions

In diabetes, pancreatic β-cells are locked in a vicious cycle, in which an impaired insulin response to glucose produces hyperglycemia, which makes β-cells more inefficient at insulin secretion, and an improvement in hyperglycemia results in at least a partial recovery of β-cell function [5]. As described in this review, hypoxia makes β-cells susceptible to dysfunction and failure, and the inhibition of HIF-1α activity and the suppression of BHLHE40 expression improve insulin secretion and hyperglycemia in animal models of diabetes, suggesting that hypoxia might be a novel target for the treatment of type 2 diabetes and that improving hypoxia might be beneficial for preventing progressive β-cell dysfunction in type 2 diabetes. However, hypoxia also induces ATF3 expression, AMPK activation, UPR^ER^ inactivation, and ROS production (Figure 5). Furthermore, HIF-1α mainly drives the response to acute hypoxia, and its expression decreases during prolonged hypoxia [70,71]. Therefore, the severity and duration of hypoxia might differentially activate adaptive responses in β-cells. A more in-depth characterization of the relative contribution of each adaptive pathway in the process of β-cell hypoxia is necessary to deepen our understanding of the pathophysiology of type 2 diabetes. Further work will provide new knowledge about the impact of hypoxic stress in β-cell dysfunction, as well as the effectiveness of β-cell hypoxia as an anti-diabetic therapeutic target.

## Figures and Tables

**Figure 1 ijms-25-04186-f001:**
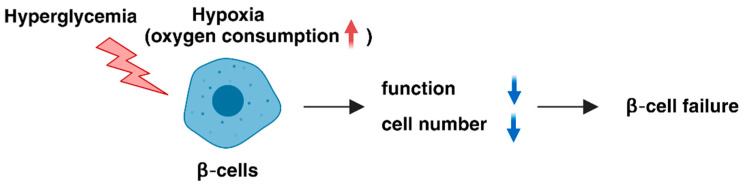
Role of hypoxic stress in pancreatic β-cells. Hyperglycemia induces hypoxia in β-cells, mostly due to the high levels of oxygen consumption required for insulin secretion. Hypoxia, in turn, exerts deleterious effects on β-cell function and number, leading to progressive β-cell failure in type 2 diabetes.

**Figure 2 ijms-25-04186-f002:**
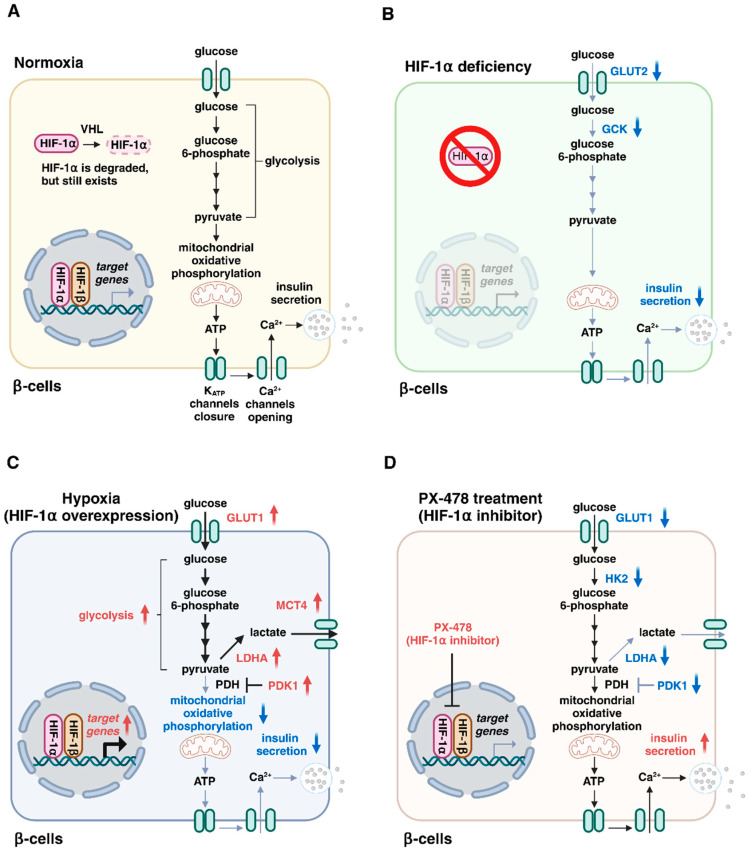
Roles of hypoxia-inducible factor (HIF)-1 in insulin secretion by β-cells. (**A**) Glucose is metabolized via the glycolytic pathway and mitochondrial oxidative phosphorylation, resulting in the generation of adenosine triphosphate (ATP), K_ATP_ channel closure, Ca^2+^ entry, and insulin exocytosis. Under normoxic conditions, HIF-1α is degraded by von Hippel–Lindau (VHL) proteins. (**B**) HIF-1α is degraded under normal oxygen conditions, but remains present in normoxic β-cells. HIF-1α deficiency causes impaired insulin secretion with a decreased expression of glucose transporter type 2 (GLUT2) and glucokinase (GCK). (**C**) HIF-1α overexpression switches glucose metabolism from mitochondrial oxidation to glycolysis, thereby leading to the attenuation of mitochondrial activity and impaired insulin secretion. (**D**) Treatment with the HIF-1α inhibitor PX-478 prevents the upregulation of HIF-1α targets (GLUT1, HK2, LDHA, and PDK1) and restores insulin secretion in metabolic workload. HK2, hexokinase 2; LDHA, lactate dehydrogenase A; MCT4, monocarboxylate transporter 4; PDH, pyruvate dehydrogenase; PDK1, pyruvate dehydrogenase kinase 1.

**Figure 3 ijms-25-04186-f003:**
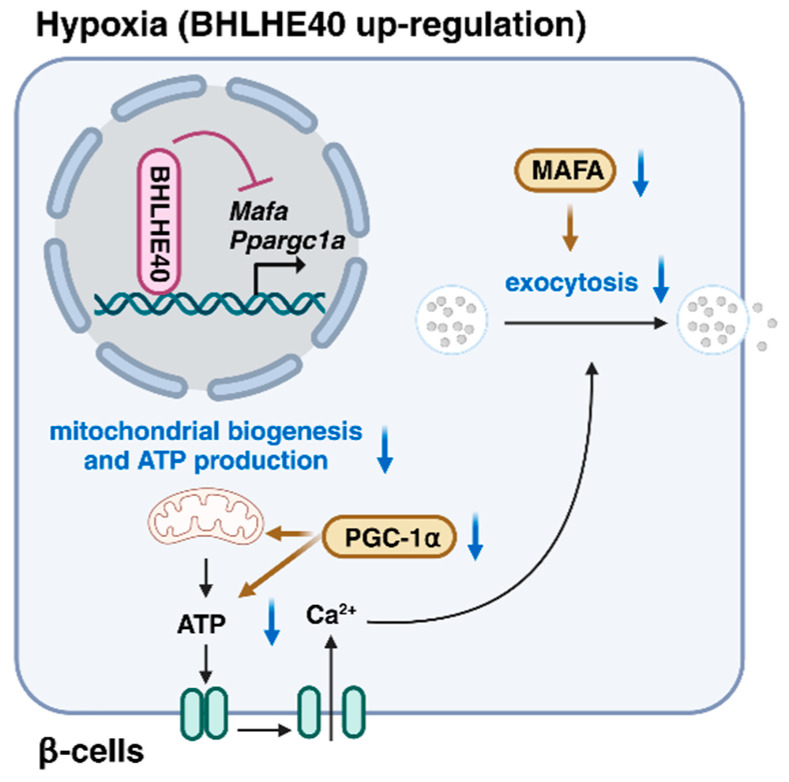
The transcriptional repressor basic helix-loop-helix family member E40 (BHLHE40) is highly induced in hypoxic β-cells. BHLHE40 inhibits insulin secretion by suppressing the expression of musculoaponeurotic fibrosarcoma oncogene family A (MAFA), a transcription factor that regulates insulin exocytosis, and peroxisome proliferator-activated receptor-γ coactivator 1α (PGC-1α), which plays important roles in mitochondrial biogenesis and adenosine triphosphate (ATP) production.

**Figure 4 ijms-25-04186-f004:**
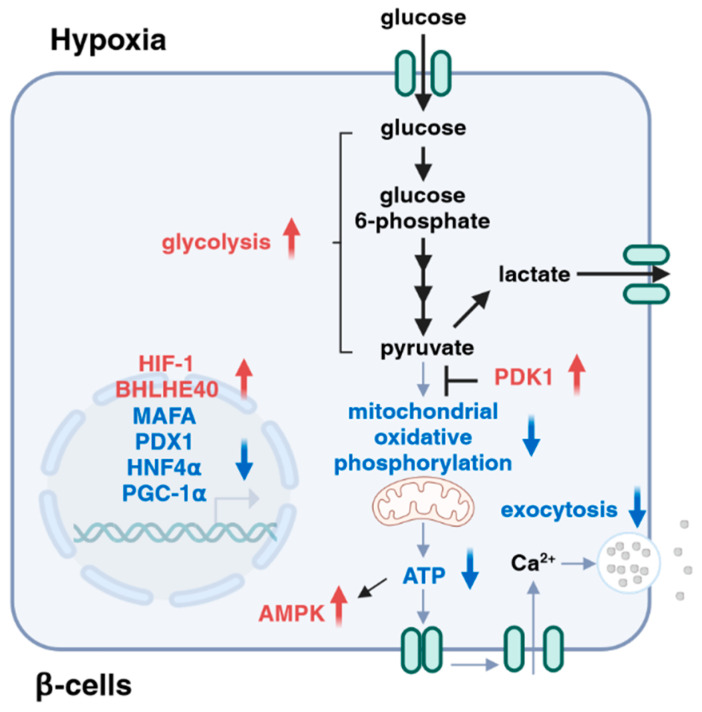
Roles of hypoxia in insulin secretion. Hypoxia affects multiple steps during the processes of glucose-stimulated insulin secretion, including dysregulation of transcription factors (e.g., MAFA, PDX1, and HNF4α), attenuation of mitochondrial activities, activation of AMPK, and inhibition of exocytosis.

**Figure 5 ijms-25-04186-f005:**
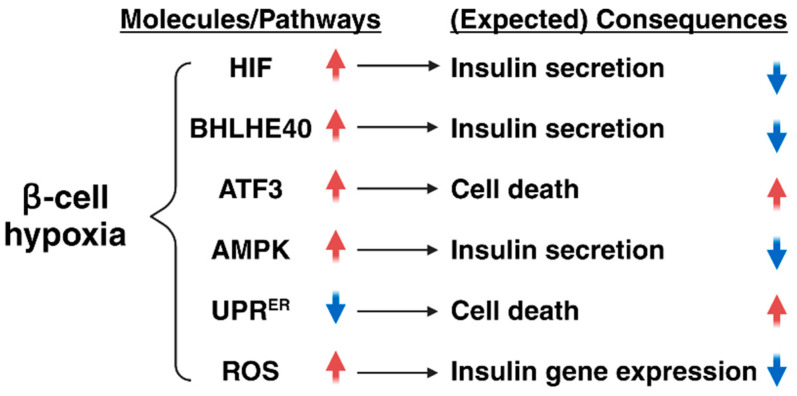
Roles of hypoxia in β-cell function and number. Hypoxia causes impaired insulin secretion through the induction of hypoxia-inducible factor 1 (HIF-1) and basic helix-loop-helix family member E40 (BHLHE40). Hypoxia also suppresses insulin secretion through the activation of adenosine monophosphate-activated protein kinase (AMPK) and the induction of reactive oxygen species (ROS), whereas, it promotes β-cell death via the induction of activating transcription factor 3 (ATF3) and the inhibition of the endoplasmic reticulum unfolded protein response (UPR^ER^).

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
