# Peer review of "Roles of β-Cell Hypoxia in the Progression of Type 2 Diabetes"

_ijms, 2024, doi:10.3390/ijms25084186_

Round 1

Reviewer 1 Report

Comments and Suggestions for Authors

This manuscript is written well and authors provided interesting information. However, authors can improve the quality of manuscript by including more information. Following are my concerns:

1.) Line 22-23: "It is estimated that 529 million people were 22 living with diabetes globally in 2021."

Please provide recent information till 2023-2024.

2.) Please provide a brief information regarding role of genes and the environment in the development of type 2 diabetes.

3.) Please explain briefly Insulin secretion in normoxia and hypoxia

4.) Can authors provide in brief role of hypoxia in Beta cell growth.

5.) That would be great if authors can explain the clinical significance of this information.

Comments on the Quality of English Language

Minor editing of English language required.

Author Response

We are grateful for the reviewer’s comment that “This manuscript is written well and authors provided interesting information” and for the constructive suggestions. We have revised the manuscript based on these valuable comments.

  • Line 22-23: "It is estimated that 529 million people were 22 living with diabetes globally in 2021." Please provide recent information till 2023-2024

We thank the reviewer for pointing this out. To our knowledge, the statistics of 2021 are most recent official data [Ong et al. 2023]. Because new estimates of the number of diabetic patients are published in that paper, we have added the following text to the revised manuscript: “… the number of people with diabetes is projected to more than double to 1.3 billion people worldwide by 2050” (lines 24-25 in the revised manuscript).

  • Please provide a brief information regarding role of genes and the environment in the development of type 2 diabetes.

We thank the reviewer for pointing this out. Based on the reviewer’s comment, we have added the following sentence to the revised manuscript: “Type 2 diabetes results from the complex interplay of multiple genetic and environmental factors. The genetic background causes insulin resistance and β-cell dysfunction, while overweight and physical inactivity exacerbate these metabolic abnormalities [DeFronzo et al. 2011; McCarthy 2010; Ling et al. 2019]” (lines 25 to 28 in the revised manuscript).

  • Please explain briefly Insulin secretion in normoxia and hypoxia

In the original manuscript, we described this as follows: “In pancreatic β-cells, glucose is metabolized into pyruvate via glycolysis, and is further oxidized in the mitochondria to produce adenosine triphosphate (ATP) by oxidative phosphorylation. An increase of ATP closes ATP-sensitive potassium channels, leading to membrane depolarization, calcium influx, and the exocytosis of insulin [14]” (lines 39 to 42 in the original manuscript). In the revised manuscript, we have changed the beginning to “In normoxic pancreatic β-cells, …” (line 43 in the revised manuscript).

In addition, based on the reviewer’s comment, we have added the following text to the revised manuscript: “From the above, hypoxia affects multiple steps during the processes of glucose-stimulated insulin secretion, Specifically, hypoxia attenuates insulin secretion by shifting glucose metabolism from mitochondrial respiration to glycolysis through the activation of HIF-1. Hypoxia also inhibits insulin secretion by suppressing the expression of MAFA (exocytosis) and PGC-1α (ATP production) through the activation of transcriptional repressor BHLHE40. In addition, hypoxia-induced activation of AMPK downregulates the expression of HNF4α, leading to defective insulin secretion. Furthermore, hypoxia-induced ROS production might inhibit insulin gene expression through the dysregulation of PDX1.” (lines 217 to 225 in the revised manuscript). We have also added the new Figure 4 to explain insulin secretion under hypoxic conditions.

  • Can authors provide in brief role of hypoxia in Beta cell growth.

We previously reported that hypoxia inhibits β-cell growth (Tsuyama et al. 2023). We have added the following information to the revised manuscript: “Consistently, exposure of MIN6 β-cells to 5% oxygen tension causes cellular hypoxia with impaired insulin secretion and inhibits β-cell growth …” (lines 59 to 61 in the revised manuscript).

  • That would be great if authors can explain the clinical significance of this information.

We thank the reviewer for pointing this out. Based on the reviewer’s suggestion, we have added the following text to the revised manuscript: “As described in this review, hypoxia makes β-cells susceptible to dysfunction and failure, and inhibition of HIF-1α activity and suppression of BHLHE40 expression improve insulin secretion and hyperglycemia in animal models of diabetes, suggesting that hypoxia might be a novel target for the treatment of type 2 diabetes and that improving hypoxia might be beneficial for preventing progressive β-cell dysfunction in type 2 diabetes.” (lines 234 to 239 in the revised manuscript).

References

DeFronzo, R.A.; Abdul-Ghani, M.A. Preservation of β-Cell Function: The Key to Diabetes Prevention. Journal of Clinical Endocrinology and Metabolism 2011, 96, 2354-66.

Ling, C. et al. Epigenetics in Human Obesity and Type 2 Diabetes. Cell Metab 2019, 29, 1028–1044.

McCarthy, M.I. Genomics, Type 2 Diabetes, and Obesity. New England Journal of Medicine 2010, 363, 2339–2350.

Ong, K.L.et al. Global, Regional, and National Burden of Diabetes from 1990 to 2021, with Projections of Prevalence to 2050: A Systematic Analysis for the Global Burden of Disease Study 2021. The Lancet 2023, 402, 203-234.

Tsuyama, T.; Sato, Y.; Yoshizawa, T.; Matsuoka, T.; Yamagata, K. Hypoxia Causes Pancreatic b‐cell Dysfunction and Impairs Insulin Secretion by Activating the Transcriptional Repressor BHLHE40. EMBO Rep 2023, 24.

Reviewer 2 Report

Comments and Suggestions for Authors

The authors present a review article that focuses on advances in the contribution of β-cell hypoxia to the development of β-cell dysfunction in type 2 diabetes.

explore advancements in allergen immunotherapy for the treatment of atopic dermatitis.

I would like to raise the following concerns.

1.This review article provides clear information on the contribution of β-cell hypoxia to the development of β-cell dysfunction in type 2 diabetes.

2.If this is a systematic review, it may require presenting a PRISMA flowchart.

Author Response

We are grateful for the reviewer’s comment that “This review article provides clear information on the contribution of β-cell hypoxia to the development of β-cell dysfunction in type 2 diabetes” and for the helpful suggestions.

  • This review article provides clear information on the contribution of β-cell hypoxia to the development of β-cell dysfunction in type 2 diabetes.

We thank the reviewer for his/her very kind comments about the manuscript.

  • “If this is a systematic review, it may require presenting a PRISMA flowchart.

We thank the reviewer for pointing this out. We tried to highlight the impact of β-cell hypoxia on the development of type 2 diabetes, and searched for relevant papers in PubMed using terms such as “β-cell,” “hypoxia,” and “insulin secretion.” However, we did not adopt the approach of systematic reviews in choosing papers.